# Attention-Shared Multi-Agent Actor–Critic-Based Deep Reinforcement Learning Approach for Mobile Charging Dynamic Scheduling in Wireless Rechargeable Sensor Networks

**DOI:** 10.3390/e24070965

**Published:** 2022-07-12

**Authors:** Chengpeng Jiang, Ziyang Wang, Shuai Chen, Jinglin Li, Haoran Wang, Jinwei Xiang, Wendong Xiao

**Affiliations:** 1School of Automation and Electrical Engineering, University of Science and Technology Beijing, Beijing 100083, China; jcp09868@163.com (C.J.); sgeccs@126.com (S.C.); joinringustb@163.com (J.L.); me_hr_926@163.com (H.W.); jinwei_xiang@126.com (J.X.); 2Beijing Engineering Research Center of Industrial Spectrum Imaging, Beijing 100083, China; 3Department of Automation, Tsinghua University, Beijing 100084, China; wangziyanghq@tsinghua.edu.cn

**Keywords:** wireless rechargeable sensor network, deep reinforcement learning, multi-agent, attention-shared, mobile charging

## Abstract

The breakthrough of wireless energy transmission (WET) technology has greatly promoted the wireless rechargeable sensor networks (WRSNs). A promising method to overcome the energy constraint problem in WRSNs is mobile charging by employing a mobile charger to charge sensors via WET. Recently, more and more studies have been conducted for mobile charging scheduling under dynamic charging environments, ignoring the consideration of the joint charging sequence scheduling and charging ratio control (JSSRC) optimal design. This paper will propose a novel attention-shared multi-agent actor–critic-based deep reinforcement learning approach for JSSRC (AMADRL-JSSRC). In AMADRL-JSSRC, we employ two heterogeneous agents named charging sequence scheduler and charging ratio controller with an independent actor network and critic network. Meanwhile, we design the reward function for them, respectively, by considering the tour length and the number of dead sensors. The AMADRL-JSSRC trains decentralized policies in multi-agent environments, using a centralized computing critic network to share an attention mechanism, and it selects relevant policy information for each agent at every charging decision. Simulation results demonstrate that the proposed AMADRL-JSSRC can efficiently prolong the lifetime of the network and reduce the number of death sensors compared with the baseline algorithms.

## 1. Introduction

Wireless sensor networks (WSNs) have been widely applied in target tracking, environment monitoring, intelligent medical and military monitoring, etc. [1,2], which have advantages including fast construction, self-organization, fault tolerance, and low-cost deployment [3]. Meanwhile, WSNs are usually composed of a large scale of sensors deployed in an area. However, sensors in WSN are always powered by batteries, and the capacity of these batteries is constrained by the volume of the sensor, which limits the lifetime of the sensors. Furthermore, the energy constraint problem affects the quality of service of WSN directly and greatly hinders the development of WSN. In recent years, the breakthrough of wireless energy transmission (WET) technology has greatly promoted the wireless rechargeable sensor networks (WRSNs) [4], since it provides a highly reliable and efficient energy supplement for the sensors. Particularly, a promising method to overcome the energy constraint problem in WRSNs is mobile charging by employing one or more mobile chargers (MCs) with a high capacity to charge sensors via WET. The MC can move to sensors autonomously and charge them according to a mobile charging scheduling scheme, which is formulated by MC based on the status information of sensors, including the residual energy, energy consumption rate, and position of sensors in WRSN. The status information of sensors is highly controllable and predictable. Theoretically, WRSNs could work indefinitely under a well-designed charging scheme [5]. Therefore, the design of the charging scheme in WRSN is critical, and it has drawn extensive attention from the research community.

There are plenty of works that have been presented to design the mobile charging schemes on WRSN. According to whether MC carries the determined charging scheme before starting from the base station, the existing works can be divided into two categories [6,7,8,9,10,11,12,13,14,15,16]: (1) offline methods [7,8,9,10,11,12] and (2) online methods [6,13,14,15,16]. In offline methods, before starting from the base station, MC will formulate a transparent charging scheme according to the status of sensors, including accurate location, fixed energy consumption rate, regular information transmission rate, etc. The MC will charge sensors with a scheduled trajectory determined by the charging scheme. The offline methods ignore the dynamic change in the status of sensors. Hence, the offline method is not suitable for dealing with application scenarios where the energy consumption rate of the sensor changes in real time and the large-scale WRSN. For example, Yan et al. [17] first attempted to introduce particle swarm optimization into optical wireless sensor networks, which could optimize the positioning of nodes, reduce the energy consumption of nodes effectively and converge faster. In [18], Shu et al. made the first attempt to deal with the jointly charging energy and designing operation scheduling in WRSN. They proposed an f-Approximate algorithm to address this problem and verify that the proposed algorithm could obtain an average 39.2% improvement of network lifetime beyond the baseline approaches. In [19], Feng et al. designed a novel algorithm called the newborn particle swarm optimization algorithm for charging-scheduling in industrial rechargeable sensor networks by adding new particles to improve the particle diversity. This improvement made the algorithm achieve better global optimization ability and improved the searching speed. V.K. Chawra et al. proposed a novel algorithm for scheduling multiple mobile rechargers using the hybrid meta-heuristic technique in [20], which combined the best features of the Cuckoo Search and Genetic Algorithm to optimize the path scheduling problem to achieve shorter charging latency and more significant energy usage efficiency. To enhance the charging efficiency [21,22,23,24], Zhang et al., Liang et al., and Wu et al. proposed some hierarchical charging methods for multiple MCs to charge sensors and themselves.

Different from the offline methods, in some application scenarios, the energy consumption rate of sensors is time-variant, and there are many uncertain factors in the network, which make the offline approaches unable to obtain an acceptable charging scheduling scheme according to the information in the network, while online approaches could successfully deal with these issues. The specific implementation is that the MC does not need to know the status of sensors clearly before starting from the base station but only needs to build candidate charging queues. When the residual energy of the sensor is lower than the set threshold, it will send a charging request and its energy information to the MC. The MC accepts the charging request and inserts it into all candidate charging queues. Then, the charging sequence will update according to the status of the sensors. For example, Lin et al. aimed to maximize the charging efficiency while minimizing the number of dead sensors to achieve the purpose of prolonging the lifetime of WRSN in [16]. Therefore, they developed a temporal–spatial real-time charging scheduling algorithm (TSCA) for the on-demand charging architecture. Furthermore, they also verified that the TSCA algorithm could obtain a better charging throughput, charging efficiency, and successful charging rate than the existing online algorithms, including Nearest-Job-Next with Preemption scheme and Double Warning thresholds with Double Preemption charging scheme. Feng et al. [25] proposed a mobile energy charging scheme that can improve the charging performance in WRSN by merging the advantages of online mode and offline mode. It includes the dynamicity of sensors’ energy consumption in the online mode and the benefit of lower charging consumption by optimizing the charging path of the mobile charger in offline mode. Kaswan et al. converted a charging scheduling problem to a linear programming problem and presented a gravitational search algorithm [26]. This approach presented a novel agent representation scheme and an efficient fitness function. In [27], Tomar et al. proposed a novel scheduling scheme for on-demand charging in WRSNs to address the joint consideration of multiple mobile chargers and the issue of ill-timed charging response to the nodes with variable energy consumption rates.

Unfortunately, although the online methods can address the mobile charging dynamic scheduling problem, they still have disadvantages, including short-sightedness, non-global optimization, and unfairness. Specifically, most recent works assume that the sensor closest to MC is usually inserted into the current charging queue. Meanwhile, sensors with low energy consumption rates are always ignored, resulting in their premature death and a reduction in the service quality of the WRSN. It is generally known that the mobile charging path planning problem in WRSN is a Markov decision process, which has been proved to be an NP-hard problem in [28]. Therefore, the most difficult problem is how to design an effective scheduling scheme to find the optimal or near-optimal solution more quickly and reliably when the size of network increases gradually. 

It is known that Reinforcement Learning (RL) is an effective method to address the Markov decision process. As mentioned above, the charging scheduling problem in WRSN is NP-hard; thus, it is unable to provide available optimal labels for supervised learning. However, the quality of a set of charging decision can be evaluated via the reward feedback. Therefore, we need to design a reasonable reward function according to the states of WRSN for RL. During the interaction between agent and environment, the charging scheduling scheme will be found through learning strategies that can maximize the reward. There are several works that have tried to solve the charging scheduling problem with RL algorithms. For example, Wei et al. [29] and Soni and Shrivastava [30] proposed a charging path planning algorithm (CSRL), combining RL and MC to extend the network lifetime and improve the autonomy of MC. However, the proposed CSRL method only suits offline mode, where the energy consumption of sensor nodes is time-invariant. Meanwhile, this method can only be used to address small-scale networks, since the Q-learning algorithm generally fails to handle high-dimensional state space or large state space. Cao et al. [28] proposed a deep reinforcement learning-based on-demand charging algorithm to maximize the sum of rewards collected by the mobile charger in WRSN, which is subject to the energy capacity constraint on the mobile charger and the charging times of all sensor nodes. A novel charging scheme for dynamic WRSNs based on an actor–critic reinforcement learning algorithm was proposed by Yang et al. [31], which aimed to maximize the charging efficiency while minimizing the number of dead sensors to prolong the network lifetime. The above works have made significant model innovation and algorithm innovation, yet they ignore the impact of sensor charging energy on the optimization performance. Although Yang et al. [31] proposed a charging coefficient to constrain the upper charging energy threshold, they assumed that all sensors have a fixed charging coefficient during the scheduling, which cannot adjust according to the needs of the sensors. Specifically, the charging coefficient could directly determine the charging energy for the sensor. Therefore, how to select the next sensor to be charged and determining its corresponding charging energy brings novel challenges to the design of the charging scheme. 

We study a joint mobile charging sequence scheduling and charging ratio control problem (JSSRC) to address the challenges mentioned above, where charging ratio is a parameter introduced to determine the charging energy for the sensor and replace on-demand charging requests with real-time changing demands. JSSRC provides timely, reliable, and global charging schemes for WRSNs in which sensors’ energy changes dynamically. Meanwhile, we propose the attention-shared multi-agent actor–critic deep reinforcement learning approach for JSSRC; this approach is abbreviated as AMADRL-JSSRC. We assume that the network deployment scenarios are friendly, barrier-free, and accessible. The transmission of information about real-time changes in energy consumption is reliable and deterministic. When the residual energy of MC is insufficient, it is allowed to return to the depot to renew its battery.

Table 1 highlights the performance comparison of the existing approaches and the proposed approach with respect to four key attributes.

The main contributions of this work are summarized as follows. (1)Different from the existing works, we consider both charging sequence and charging ratio optimization simultaneously in this paper. We introduce two heterogeneous agents named charging sequence scheduler and charging ratio controller. These two agents give the charging decisions separately under the dynamic changing environments, which aims to prolong the lifetime of the network and minimize the number of dead sensors.(2)We design a novel reward function with a penalty coefficient by comprehensively considering the tour length of MC and the number of dead sensors for AMADRL-JSSRC, so as to promote the agents to make better decisions.(3)We introduce the attention shared mechanism in AMADRL-JSSRC to the problem that charging sequence and charging ratio have different contributions to the reward function.

The rest of the paper is organized as follows: Section 2 describes the system models of WRSN and formulates the JSSRC problem. The proposed AMADRL-JSSRC approach is described in Section 3. Simulation results are reported in Section 4. The impacts of the parameters on the charging performance are discussed in Section 5. Conclusions and future work are given in Section 5.

## 2. System Model and Problem Formulation

In this section, we present the network structure, energy consumption model of sensors, energy analysis of MC, and the formulation of the charging scheduling problem in WRSNs.

### 2.1. Network Structure

In Figure 1, a WRSN with *n* heterogeneous isomorphic sensors $SN=\left\{sn^{1},sn^{2},\ldots ,sn^{n}\right\}$, an MC, a base station (BS), and a depot are adopted. It is assumed that due to different information transmission tasks, all sensors have the same energy capacity $E^{sn}$ and sensing ability but different energy consumption rates. They are deployed in a 2D area without obstacles; the positions of all sensors are fixed and can be determined accurately, and they are recorded as $\left(x_{i},y_{i}\right),i\in \left[1,n\right]$, and the position of *BS* is set as $\left(x_{0},y_{0}\right)$. Therefore, a weighted undirected graph $G=\left(\left\{BS,SN\right\},D^{sn},E^{0},E^{c}\right)$ is used to describe the network model of WRSN, where $D^{sn}$ is the set of distances between sensors, which is expressed as $D^{sn}=\left\{d_{ij}|d_{ij}=d\left(sn^{i},sn^{j}\right)\right\},i,j\in \left[1,n\right]$ with $d\left(sn^{i},sn^{j}\right)=\sqrt{{\left(x_{i}-x_{j}\right)}^{2}+{\left(y_{i}-y_{j}\right)}^{2}}$. The set of initial residual energy and the energy consumption rate of each sensor are represented by $E^{0}$ and $E^{c}$, respectively. $\left(x_{D},y_{D}\right)$ is defined as the position of the depot.

It is assumed that each sensor in WRSN collects data and communicates with *BS* via ad hoc communication. The BS could estimate their residual energy according to data sampling frequency and transmission flow. MC can obtain the state information of the sensor but will not interfere with the working state of the sensor. Meanwhile, the total moving distance of MC during the charging tour is defined as Dis.

Although, in theory, the lifetime of the network can be extended indefinitely with single or multiple MC. The network will shut down, since the energy modules of sensors will age. Therefore, inspired by [28,31], we define the lifetime in this article as below.

**Definition** **1** **(Lifetime).**
*The lifetime of WRSNs is defined as the period from the beginning of the network to the number of dead sensors reaching a threshold.*


The lifetime and the threshold are described with $T_{life}$ and ω%, respectively. Furthermore, the abbreviations used in this paper are summarized in Table 2.

### 2.2. Energy Consumption Model of Sensors

The energy of the sensor is mainly consumed in data transmission and reception. Therefore, based on [32,33], the energy consumption model at time slot *t* is adopted as below:(1)$$e_{c}^{i}\left(t\right)=\rho \sum_{k=1,k\neq i}^{n}f_{k,i}^{r}\left(t\right)+\sum_{j=1,j\neq i}^{n}[\varsigma_{i,j}^{t}\cdot f_{i,j}^{t}\left(t\right)+\varsigma_{i,B}^{t}\cdot f_{i,B}^{t}\left(t\right)]$$
where ρ is the energy consumption for receiving or transmitting 1 kb data from sensor $sn^{i}$ to sensor $sn^{j}$ (or BS). $\varsigma_{i,j}^{t}=\xi_{1}+\xi_{2}\cdot d_{i,j}^{r}$ represents the energy consumption for transmitting 1 kb data between each sensor, where $d_{i,j}$ is the distance between $sn^{i}$ and $sn^{j}$. $\xi_{1}$ and $\xi_{2}$ represent the distance-free and distance-related energy consumption index, respectively. *r* is the signal attenuation coefficient. $f_{k,i}^{r}$ means the data flow of receiving, $f_{i,j}^{t}\left(1\leq j\leq n\right)$ and $f_{i,B}^{t}$ are the data flow of transmitting from $sn^{i}$ to $sn^{j}$ and BS. Hence, $\rho \sum_{k=1,k\neq i}^{n}f_{k,i}^{r}\left(t\right)$ represents the energy consumption of $sn^{i}$ receiving information from all sensor nodes. $\sum_{j=1,j\neq i}^{n}[\varsigma_{i,j}^{t}\cdot f_{i,j}^{t}\left(t\right)+\varsigma_{i,B}^{t}\cdot f_{i,B}^{t}\left(t\right)]$ is the energy consumption of $sn^{i}$ by sending information to other sensors and BS.

### 2.3. Charging Model of MC

In this paper, the sensors in WRSN are charged by MC wirelessly, and the empirical wireless charging model is defined as [34]
(2)$$P_{c}=\frac{G_{s}G_{r}\eta }{L_{p}}{\left(\frac{\lambda }{4\pi \left(d_{ms}+\beta \right)}\right)}^{2}P_{0}$$
where $d_{ms}$ represents the distance between the sensor and the mobile charger, $P_{0}$ is the output power, $G_{s}$ is the gain of the source antenna which is equipped on the mobile charger, $G_{r}$ is the gain of the receiver antenna, $d_{ms}$ is the distance between the mobile charger and the sensor, $L_{p}$, and λ denote the rectifier efficiency and the parameter to adjust the *Friis’* free space equation for short-distance transmission, respectively.

Since the MC moves to the position near the sensors, the distance can be regarded as a constant. Therefore, (2) can be simplified to (3)
(3)$$P_{c}=\frac{\Delta }{\mu }\cdot P_{0},$$
in which $\Delta =G_{s}G_{r}\eta \lambda^{2}/16\pi^{2}L_{p}$, $\mu ={\left(d_{ms}+\beta \right)}^{2}$. 

The moving speed of the MC is set as $v_{ms}$, and the energy consumed per meter is $e_{m}J$. The capacity of MC is $E^{mc}$, and the target sensor will be charged with one-to-one charging mode only when the MC reaches it.

### 2.4. Problem Formulation

We define three labels to describe the working states of the visited point at time slot *t*, i∈[0,n], i=visit,i≠visit and i=dead. They represent that $sn^{i}$ is selected to charge, not be selected and dead, respectively, while i=0 represents that the visited point is a depot. The residual energy of the sensor is defined as $e_{r}^{i}\left(t\right)$, the charging demand of sensor $sn^{i}$ is defined as $e_{d}^{i}\left(t\right)$ and the residual energy of MC is defined as $e_{mc}^{r}\left(t\right)$.

At time slot *t*, the residual energy of the sensor is described as (4), and the charging demand will also be updated with (5)
(4)$$e_{r}^{i}\left(t\right)=\{\begin{array}{cc} e_{r}^{i}\left(t-1\right)-e_{c}^{i}\left(t\right) & i\neq visit,i\in \left[1,n\right]\\ e_{r}^{i}\left(t-1\right)+P_{c} & i=visit,\;i\in \left[1,n\right]\\ 0 & i=dead,\;i\in \left[1,n\right] \end{array}$$
(5)$$e_{d}^{i}\left(t\right)=\varepsilon E^{sn}-e_{r}^{i}\left(t\right)$$
where ε is the charging ratio, it could decide the upper threshold of charging energy, and its value range in (0,1].

To effectively charge the sensors, more energy in the MC should be used on charging sensors, while the energy wasted on moving between the sensors and the depot should be minimized. Hence, within the network lifetime $T_{life}$, the JSSRC problem under WRSNs with the dynamic energy changing is defined as below.

**Definition** **2** **(JSSRC).***The joint mobile charging sequence scheduling and the charging ratio control problem, which aims to prolong the lifetime of the network and minimize the number of dead sensors in WRSNs with dynamic energy changing, is defined as the JSSRC problem*.

The relevant notations are defined as follows: at time slot t, the current state of sensor *i* is defined as (6) according to its residual energy, if $\tau_{i}\left(t\right)=1$, it indicates that the sensor is alive, and $\tau_{i}\left(t\right)=0$ represents that the sensor has died.
(6)$$\tau_{i}\left(t\right)=\{\begin{array}{cc} 1, & e_{r}^{i}\left(t\right)>0\\ 0, & e_{r}^{i}\left(t\right)\leq 0 \end{array}$$

Furthermore, the number of dead sensors is defined as $N_{d}$(t), which is obtained with (7)
(7)$$N_{d}\left(t\right)=n-\sum_{i=1}^{n}\tau_{i}\left(t\right).$$

There are three termination conditions of the JSSRC scheme, and they are described with (8):(1)The number of dead sensors reaches ω% of the total number, ω∈(0,100].(2)The remaining energy of MC is insufficient to return to the depot.(3)The target lifetime or the base time is reached. (8)$$\begin{array}{c} N_{d}\left(t\right)=\omega \cdot n\\ e_{mc}^{r}\left(t\right)<d^{\prime }\cdot e_{m}\\ t=T_{target} \end{array}$$
where $d^{\prime }$ represent the distance from the MC’s current location to the depot, t is the running time of the test, and $T_{target}$ is a given base time. Specifically, when any of the termination conditions in (8) are met, the charging process will end.

Then, within the network lifetime, the JSSRC problem can be formulated as
(9)$$\begin{array}{cc} min & Dis\\ min & N_{d}\\ s.t. & \left(4\right),\left(5\right),\left(6\right),\left(7\right),\left(8\right) \end{array}$$

## 3. Details of the Attention-Shared Multi-Agent Actor-Critic Based Deep Reinforcement Learning Approach for JSSRC (AMADRL-JSSRC)

JSSRC is a joint scheduling problem with sequence scheduling and charging ratio control; it is difficult to schedule them simultaneously with the traditional single-agent reinforcement learning algorithm. Therefore, the multi-agent reinforcement learning algorithm is introduced to solve this problem. In this section, we first briefly introduce multi-agent reinforcement learning algorithms. Then, we model the provided problem and propose the AMADRL-JSSRC.

### 3.1. Basis of Multi-Agent Reinforcement Learning

Multi-agent reinforcement learning is developed on the basis of the reinforcement learning algorithm, which is often described as the Markov game (or stochastic game) [35]. Multi-agent reinforcement learning is also an important branch of machine learning and deep learning, which aims to improve the shortcomings of multi-objective control that cannot be achieved by a single agent. Each agent can be a cooperative, competitive or mixed relationship, and they learn how to make decisions in an environment by observing the rewards obtained after the environment performs some actions. Specifically, there are *m* agents; each agent first receives their own observations $o_{\vartheta }\left(\vartheta \in \left[1,m\right]\right)$. Then, we select an operation $a_{\vartheta }$ from action spaces, which are subsequently sent to the environment. After that, the environment state transits from S to $S^{\prime }$, and each agent receives a reward $r_{\vartheta }$ associated with these transitions. The purpose of training agents is to collect accumulated rewards from multiple agents as much as possible.

### 3.2. Learning Model Construction for JSSRC

The tuple {S,A1,A2,R,S′} is used to define the JSSRC scheme, where S is the state space of two agents, A1 and A2 are the action spaces, R is the sum of rewards obtained by two agents after performing actions, and S′ is the state of the environment after executive action [36]. A state transition function is defined as T with T:S×A1×A2→P(S), which is the probability distribution over the possible next states. Furthermore, there are two agents in JSSRC with their own set of observations, O1 and O2. The environment state is defined as S=(O1,O2), and the new environment state is defined as S′=(O′1,O′2). The reward for each agent also depends on the global state and actions of all agents; thus, we have the reward function, Rϑ:S×A1×A2→ℝ, where ϑ is the number of the agent, and ℝ is the set of all possible rewards.

The time step is defined as the time slot when the scheduling decision is made. Hence, at the *k*-th time step, the MC visits position *i* and completes the charging decision, where i∈[0,n]. *K* is defined as the maximum time step when any of the termination conditions are met. The time slot corresponding to the *k*-th time step is defined as t(k); when the action of the *k*-th time step is completed, the corresponding time slot is recorded as $t\left(\underline{k}\right)$.

A scheduling example of JSSRC is shown in Figure 2. To express clearly, we omit the information communication process between sensors, leaving only the scheduling decision and the charging path. The relationship between the time slot and the time step is described in the upper part of the figure. Within the network lifetime $T_{life}$, two agents determine two actions $a_{1}\left(k\right)\;and\;a_{2}\left(k\right)$ according to their observations $o_{1}\left(k\right)$ and $o_{2}\left(k\right)$ in state S(k) at time step *k*. $a_{1}$ represents the decision decided by agent 1 to choose the next sensor to be charged, and $a_{2}$ represents the decision decided by agent 2 to control the charging ratio. Agents obtain their policy according to the continuous exploration and calculate the rewards R through the obtained strategies at the end of the $T_{life}$. Then, the states, actions, policies, and rewards of the environment are defined as follows.

**States of the environment**: The state space of the environment in JSAAC includes the state information of the MC and sensors, which are defined as $S_{mc}$ and $S_{net}$, respectively. An example of the information at time step *k*, $S_{mc}$ is $\left(p_{mc}\left(k\right),e_{mc}^{r}\left(k\right)\right)$, $S_{net}$ is $\left(p_{sn}^{i}\left(k\right),e_{d}^{i}\left(k\right),e_{c}^{i}\left(k\right)\right)$, where $p_{mc}\left(k\right)$ and $e_{mc}^{r}\left(k\right)$ are the position and the residual energy of MC, $p_{sn}^{i}\left(k\right)$ is the position of $sn^{i}$ to be visited, $e_{d}^{i}\left(k\right)$ and $e_{c}^{i}\left(k\right)$ are the charging demand and the energy consumption rate of $sn^{i}$, where k∈[0,K] and i∈[0,n]. $sn^{0}$ represents the depot, and the value of $e_{d}^{0}$ is 0 because the depot does not need to be charged. The state embedding is a 5×K dimensional vector at time step k with $S\left(k\right)=\left(S_{mc}\left(k\right),S_{net}\left(k\right)\right)$; only the position of the sensor is a static element, the others are dynamic.

**Actions of the environment**: The actions in JSSRC represent the decision of the target sensor and the charging ratio, which are determined by two agents.

**Policies of the environment**: The policy for a single agent is described with a=π(o), where a is an action, o is the observation of the agent, and π is the policy. In JSSRC, there are two agents; we define two agents with policies parameterized by $\theta =\{\theta_{1},\theta_{2}\}$ and let $\pi =\{\pi_{1},\pi_{2}\}$ with $\pi_{\vartheta }:O_{\vartheta }\mapsto P\left(A_{\vartheta }\right)$ where P(ϑ)∈[0,1],ϑ=1,2. The main goal of JSSRC is to learn a set of optimal policies to maximize two agents’ expected discounted rewards.

**Rewards of the environment**: Reward is used to evaluate the action; its value is obtained by the agent after executing an action. In this paper, our goal is to improve the charging performance of WRSN, which includes minimizing the moving distance of MC and reducing the number of dead sensors. Since the total number of dead sensors is inversely proportional to the reward, if the performed actions lead to more sensors being dead, we will give a penalty for this behavior. Therefore, the expected discounted rewards for two agents are defined with (10), and the immediate reward obtained after performing the actions at the *k*-th time step is defined with (11).

The expected discounted rewards for two agents can be defined as
(10)$$R_{\vartheta }\left(\pi_{\vartheta }\right)=E_{a_{1}\sim \pi_{1},a_{2}\sim \pi_{2},S\sim T}\left[\sum_{k=0}^{K}\gamma^{k}r_{\vartheta }\left(S\left(k\right),a_{1}\left(k\right),a_{2}\left(k\right)\right)\right],\vartheta \in \left[1,2\right].$$
(11)$$r_{\vartheta }\left(S\left(k\right),a_{1}\left(k\right),a_{2}\left(k\right)\right)=\varpi^{d\left(k-1,k\right)}+\left(-\epsilon \right)N_{d}\left(k\right).\;$$
(12)$$Dis_{a_{1}\sim \pi_{1},a_{2}\sim \pi_{2}}=\sum_{k=1}^{K}d\left(k-1,k\right)$$
where the action space of $a_{1}$ is $A_{1}=\left\{a_{1}|a_{1}\in \left\{0,1,\ldots ,n\right\}\right\}$, and the action space of $a_{2}$ is $A_{2}=\{a_{2}|a_{2}\in \left\{0.5,0.6,\ldots ,1\right\}\}$. ϖ is a reward coefficient between 0 and 1, which can ensure the shorter the moving distance is, the greater the reward that will be obtained. The $N_{d}\left(k\right)$ indicates the number of new dead sensors after the actions at the *k*-th time step are performed, and ϵ is the penalty coefficient. In (12), $Dis_{a_{1}\sim \pi_{1},a_{2}\sim \pi_{2}}$ represents the total moving distance obtained after performing the actions when the termination conditions are met. Obviously, the decision of the charging sequence and the charging ratio have different contributions to the reward function, which brings difficulties to the design of the algorithm.

**State Space Update of the environment**: One episode of the JSSRC can be formed as a finite sequence of decisions, observations, actions, and immediate rewards, which is described in Table 3.

To display the specific update process of states, we assume that the MC is located at the depot at time step 0. At each time step, MC decides the next charged sensor from *SN* and determines the corresponding charging ratio for it. It is defined that the residual energy of sensor $sn^{i}$ before charging and after charging are $e_{r}^{i}\left(k\right)$ and $e_{r}^{i}\left(\underline{k}\right),$ respectively. The charging demand of each sensor and the residual energy of MC will be updated after performing the charging operation at time step *k*. They are shown as follows:(13)$$e_{r}^{i\left(i=visit\right)}\left(k\right)=\max\left\{0,e_{r}^{i}\left(\underline{k-1}\right)-t_{m}\left(k\right)\cdot e_{c}^{i}\left(k\right)\right\}$$
(14)$$e_{r}^{i\left(i=visit\right)}\left(\underline{k}\right)=\varepsilon_{i}\left(k\right)\cdot E^{sn}$$
(15)$$e_{r}^{i\left(i\neq visit\right)}\left(k\right)=\max\left\{0,e_{r}^{i}\left(\underline{k-1}\right)-t_{m}\left(k\right)\cdot e_{c}^{i}\left(k\right)\right\}.$$
(16)$$e_{r}^{i\left(i\neq visit\right)}\left(\underline{k}\right)=e_{r}^{i\left(i\neq visit\right)}\left(k\right)-t_{c}\left(k\right)\cdot e_{c}^{i}\left(k\right).$$
where $t_{m}\left(k\right)$ is the moving duration of the MC between the *k*-1-th and *k*-th time step.

It is assumed that at the *k*-1-th time step, the MC is located at $sn^{j}$, at the *k*-th step, MC is located at $sn^{i}$. Therefore, we have $d\left(k,k-1\right)=d_{ij}$, and $t_{m}\left(k\right)$ can be obtained by (17)
(17)$$t_{m}\left(k\right)=\frac{d\left(k,k-1\right)}{v_{m}}$$

If $sn^{i}$ is alive at the *k*-th time step, the charging time is
(18)$$t_{c}^{i\left(i=visit\right)}\left(k\right)=\frac{\varepsilon_{i}\left(k\right)\cdot E^{sn}-e_{r}^{i\left(i=visit\right)}\left(k\right)}{P_{c}-e_{c}^{i}\left(k\right)}$$
where $\varepsilon_{i}\left(k\right)$ is the unique charging ratio of $sn^{i}$ at the *k*-th time step.

Therefore, the charging demands of three types of working states about $sn^{i}$ are
(19)$$e_{d}^{i\left(i=visit\right)}\left(k\right)=\varepsilon_{i}\left(k\right)\cdot E^{sn}-e_{r}^{i\left(i=visit\right)}\left(k\right)$$
(20)$$e_{d}^{i\left(i\neq visit\right)}\left(k\right)=\varepsilon_{i}\left(k\right)\cdot E^{sn}-e_{r}^{i\left(i\neq visit\right)}\left(\underline{k}\right).$$
(21)$$e_{d}^{i\left(i=dead\right)}\left(k\right)=0$$

The residual energy of the MC before and after performing the charging operation is defined as $e_{r}^{mc}\left(k\right)$ and $e_{r}^{mc}\left(\underline{k}\right)$, respectively; they will update with (22) and (23)
(22)$$e_{r}^{mc}\left(k\right)=\max\left\{0,e_{r}^{mc}\left(\underline{k-1}\right)-d\left(k-1,k\right)\cdot e_{m}\right\}.$$
(23)$$e_{r}^{mc}\left(\underline{k}\right)=\max\left\{0,e_{r}^{mc}\left(k\right)-e_{d}^{i\left(i=visit\right)}\left(k\right)\right\}$$

To speed up the training and obtain feasible solutions, we give the following constraints:(1)The MC could visit any position in the network as long as its residual energy could satisfy the charging demand of the next selected sensor or is enough to move back to the depot.(2)All sensors with a charging demand greater than 0 have a certain probability of being selected as the next one to be charged.(3)The MC does not charge the sensors whose charging demands are zero.(4)If the residual energy of MC does not satisfy the charging demand of the next selected sensor, but it is enough to return to the depot, the MC is allowed to return to the depot to charge itself, and the charging time of the MC is ignored.(5)The charging decision of two adjacent time steps cannot be the same sensor or depot.(6)If the residual energy of the MC does not meet the charging demand for the next sensor, is not enough to return to the depot, or the preset network lifetime is reached, the charging plan will be ended no matter whether the sensors are still alive or not.

### 3.3. AMADRL-JSSRC Algorithm

As depicted in Figure 3, AMADRL-JSSRC’s implementation consists of the environment, the experience replay buffer (*D*), the mini-batch (*B*), the obtained rewards, and the different neural networks. The environment can be partially observed by each agent, where the actor and critic networks estimate the optimal control policies for the charging sequence scheduler and the charging ratio controller. The detail of training AMADRL-JSSRC is described in Algorithm 1.

Unlike the traditional methods such as MADDPG [36] and MAPPO [37], each agent receives information from other agents without discrimination and calculates the corresponding Q-value. In JSSRC, the contribution of the charging sequence scheduler and the charging ratio controller to the Q-value are different. Compared with the charging ratio, the decision of the charging sequence has a greater impact on the reward. To calculate the Q-value function $Q_{\vartheta }^{\varphi }\left(s,a\right)$ for agent ϑ, we introduce the attention mechanism with a differentiable key-value memory model [38,39]. This kind of mechanism does not need to make any assumptions about the temporal or spatial locality of the inputs, which is more suitable to overcome the difficulty that each agent has a different action space and contributes a different reward in this article.

At each time step, the critic network in each agent will receive the observation information $s=\left(o_{1},o_{2}\right)$ and action information $a=\left(a_{1},a_{2}\right)$, for all ϑ∈[1,2]. We define the set of all agents except for ϑ as \ϑ, and we use $\hat{\vartheta }$ as the pointer to index the set. $Q_{\vartheta }^{\varphi }\left(s,a\right)$ is defined as the function of agent ϑ which is obtained by combining with the observation information, action information, and contribution from other agents:(24)$$Q_{\vartheta }^{\varphi }\left(s,a\right)=f_{\vartheta }\left(g_{\vartheta }\left(o_{\vartheta },a_{\vartheta }\right),c_{\vartheta }\right)$$
where $f_{\vartheta }$ is a two-layer multi-layer perceptron (MLP) [40], and $g_{\vartheta }$ is a one-layer MLP embedding function. $c_{\vartheta }$ is the contribution from other agents, which is a weighted sum of the value of each agent with (25)
(25)$$c_{\vartheta }=\sum_{\hat{\vartheta }\neq \vartheta }\kappa_{\hat{\vartheta }}v_{\hat{\vartheta }}=\sum_{\hat{\vartheta }\neq \vartheta }\kappa_{\hat{\vartheta }}h\left(Vg_{\hat{\vartheta }}\left(o_{\hat{\vartheta }},a_{\hat{\vartheta }}\right)\right)$$

In (25), $v_{\hat{\vartheta }}$ is the embedding function of agent $\hat{\vartheta }$ encoded with an embedding function. Then, the shared matrix V is used for linear transformation. h is an element-wise nonlinearity activation function named leaky ReLu, which could retain some negative axis values to prevent all negative axis information from being lost. h is realized by (26)
(26)$$h\left(x\right)=\{\begin{array}{cc} x & x>0\\ \phi \cdot x & otherwise \end{array}$$
where ϕ is a very small constant.

The attention weight $\kappa_{\hat{\vartheta }}$ uses bilinear mapping (i.e., query-key system) to compare the embedded $e_{\hat{\vartheta }}$ with $e_{\vartheta }=g_{\vartheta }\left(o_{\vartheta },a_{\vartheta }\right)$, and it passes the similarity value between these two embedding into a SoftMax function:(27)$$\kappa_{\hat{\vartheta }}\propto \exp\left(e_{\hat{\vartheta }}^{T}W_{k}^{T}W_{q}e_{\vartheta }\right)$$
where the $e_{\vartheta }$ is transformed to a “query” with $W_{q}$, and the $e_{\hat{\vartheta }}$ is transformed to a “key” with $W_{k}$ [41].

To prevent vanishing gradients, the matching is scaled by the dimensionality of these two matrices. The multiple attention heads mechanism is introduced in AMADRL-JSSRC, each head with a separate set of parameters ($W_{k},W_{q},V$), which could give rise to an aggregated contribution from another agent to the agent i. We concatenate the contributions of all heads into a vector. The most important point is that each head could focus on a different weighted mixture of agents.

In AMADRL-JSSRC, the weights for extracting selectors, keys, and values are shared between two agents, because the multi-agent value function is essentially a multi-task regression problem. This parameter sharing in the critic network enables our method to learn effectively in an environment where the action space and reward for individual agents are different but share common observation features. The structure of the critic network and the structure of the multiple head attention mechanism are clearly shown in the left part of Figure 3.

### 3.4. Parameters Update in AMADRL-JSSRC

The parameters $\bar{\varphi }$ and $\bar{\theta }$ used in the critic networks and policies gradient will be updated, respectively, according to line 17 to line 24 and line 28 to line 32 in Algorithm 1.

Since the parameters are shared among critic networks in AMADRL-JSSRC, all critic networks are updated together to minimize a joint regression loss function:(28)$${ℒ}_{Q}\left(\varphi \right)=\sum_{\vartheta =1}^{2}E_{\left(s,a,r,s^{\prime }\right)\sim D}{\left[Q_{\vartheta }^{\varphi }\left(s,a\right)-y_{\vartheta }\right]}^{2}$$

In (28), $y_{\vartheta }$ is obtained by (29)
(29)$$y_{\vartheta }=r_{\vartheta }+\gamma E_{a^{\prime }\sim \pi_{\bar{\theta }}\left(s^{\prime }\right)}[Q_{\vartheta }^{\bar{\varphi }}\left(s^{\prime },a^{\prime }\right)-\alpha \log(\pi_{{\bar{\theta }}_{\vartheta }}(a_{\vartheta }^{\prime }|o_{\vartheta }^{\prime }))]$$

It is worth noting that $Q_{\vartheta }^{\bar{\varphi }}$ is used to estimate the action value for agent ϑ by receiving the observation information and action information from all agents. *D* is a replay buffer to store past experiences. In (29), α is a parameter that could trade off maximizing entropy and rewards.

Since the charging sequence decision has a greater impact on the expected reward than the charging ratio decision, in order to give the optimal policies objectively, we need to compare the value of a specific action to the value of the average action of the agent, with another agent fixed. We could determine whether said action will lead to an increase in expected return or whether any increase in reward is attributed to the actions of another agent. This problem is called multi-agent credit assignment. An effective solution is to introduce an advantage function [42] with a baseline that only marginalized the actions of the given agent from $Q_{\vartheta }^{\varphi }\left(s,a\right)$, and the form of this advantage function is shown below:(30)$$A_{\vartheta }\left(s,a\right)=Q_{\vartheta }^{\varphi }\left(s,a\right)-b\left(s,a_{\hat{\vartheta }}\right),$$
where
(31)$$b\left(s,a_{\hat{\vartheta }}\right)=E_{a_{\vartheta }\sim \pi_{\vartheta }\left(o_{\vartheta }\right)}\left[Q_{\vartheta }^{\varphi }\left(s,\left(a_{\vartheta },a_{\hat{\vartheta }}\right)\right)\right]$$

In (31), $b\left(s,a_{\hat{\vartheta }}\right)$ is the multi-agent baseline used to calculate the advantage function.

We calculate our baseline with the AMADRL-JSSRC algorithm in a single forward pass by outputting the expected return $Q_{\vartheta }^{\varphi }\left(s,\left(a_{\vartheta },a_{\hat{\vartheta }}\right)\right)$ for every possible action, $a_{\vartheta }\in A_{\vartheta }$. The expectation could be calculated exactly with (32)
(32)$$E_{a_{\vartheta }\sim \pi_{\vartheta }\left(o_{\vartheta }\right)}\left[Q_{\vartheta }^{\varphi }\left(s,\left(a_{\vartheta },a_{\hat{\vartheta }}\right)\right)\right]=\sum_{a_{\vartheta }^{\prime }\in A_{\vartheta }}\pi (a_{\vartheta }^{\prime }|o_{\vartheta })Q_{\vartheta }\left(s,\left(a_{\vartheta }^{\prime },a_{\hat{\vartheta }}\right)\right)$$

To achieve this goal, we make the following four adjustments:(1)We must remove $a_{i}$ from the input of $Q_{i}$ and output a value for every action.(2)We need add an observation encoder, $e_{\vartheta }=g_{\vartheta }^{s}\left(o_{\vartheta }\right)$, to replace the $e_{\vartheta }=g_{\vartheta }\left(o_{\vartheta },a_{\vartheta }\right)$ in (24) described above.(3)We also modify $f_{\vartheta }$ to output the Q-value of all possible actions rather than the single input action.(4)To avoid overgeneralization [43], we sample all actions from the current strategies of all agents to calculate the gradient estimation of agent ϑ rather than sampling the actions of other agents from the experience replay buffer such as [36,39].

**Algorithm 1** AMADRL-JSSRC1: Initialize the number of parallel environments for two agents as Np, initialize the update time of parallel operation as Tupdate, initialize the experience replay buffer with *D* and the minibatch with *B*, initialize the number of episodes as Ne, the number of steps per episode as Npe the number of critic updates as Ncu, the number of policy updates as Npu, and the number of multiple attention head as Nm, initialize the critic network Qφ, and actor network πθ with random parameters φ, θ, initialize the target network, φ¯←φ and θ¯←θ, Tupdate←02: **for** iep=1,…,Ne  **do**3: **Reset** environments, and obtain the initial oϑenv for each agent, ϑ4:  **for** k=1,…,Npe  do5:   Randomly select actions aϑenv~πϑ(⋅|oϑenv) for each agent ϑ, in each environment (env)    with greedy search strategy6:   Send actions to all parallel environments, then obtain oϑ′env and rϑenv for all agents7:   Store transitions for all environments in *D*8:   Tupdate=Tupdate+Np9:   **if** Tupdate≥min steps per update **then**10:    **for** j=1,..,Ncu do11:    Sample *B*12:    **function** Update Critic (*B*):13:     Unpack the mini-batch (*B*)14:     (o1,2B,a1,2B,r1,2B,o1,2′B)←B15:     Calculate Qiφ(o1,2B,a1,2B) for two agents in parallel16:     Calculate a1′B~π1θ(o1′B) and a2′B~π2θ(o2′B) with target policies17:     Calculate Qϑφ¯(o1,2′B,a1,2′B) for two agents in parallel with the target critic18:     Update critic with ∇ℒQ(φ) shown in (28) and Adam optimizer [44]19:    **end function** Update Critic20:    **end for**21:    **for** j=1,..,Npu do22:    Sample Nm×(o1,2)~D23:    **function** Update Policies (o1,2B)24:     Calculate a1,2B~πϑθ¯(oi′B),ϑ=1,225:     Calculate Qϑφ(o1,2B,a1,2B) for two agents in parallel26:     Update policies with ∇θϑJ(πθ) shown in (33) and Adam optimizer [44]27:     **end function** Update Policies28:   **end for**29:    Update target parameters:          φ¯=τφ¯+(1−τ)φ, θ¯=τθ¯+(1−τ)θ30:      Tupdate←031:      **end if**32:    **end for**33:  **end for**34: **Output**: The parameters of target actor 

Therefore, the policies of each agent will be updated by:(33)$$\nabla_{\theta_{\vartheta }}J\left(\pi_{\theta }\right)=E_{s\sim D,a\sim \pi }[\nabla_{\theta_{\vartheta }}\log(\pi_{\theta_{\vartheta }}(a_{\vartheta }|o_{\vartheta }))\left(-\alpha \log\left(\pi_{\theta_{\vartheta }}\left(a_{\vartheta }|o_{\vartheta }\right)\right)+Q_{\vartheta }^{\varphi }\left(s,a\right)-b\left(s,a_{\hat{\vartheta }}\right)\right)]$$

## 4. Experimental Setup and Results

In this section, we will conduct experiments to evaluate the performance of AMADRL-JSSRC. The simulations are divided into two phases: (1) the training phase of AMADRL-JSSRC and (2) the testing phase for a comparative study with baseline algorithms. The experiment setting and training details are described in Section 4.1. The testing details of the comparison with the baseline algorithms are described in Section 4.2.

### 4.1. Experimental Environment and Details

We conduct the AMADRL-JSSRC using Python 3.9.7 and TensorFlow 2.7.0 over 10,000 episodes, and each episode is divided into 100 time slots. Then, AMADRL-JSSRC has tested over ten episodes, where the average values of the important metrics are calculated.

We use the same simulation settings as described in [31], and some details are supplemented here. We assume that the locations of sensors are assigned uniformly at random in the unit square [0,1]×[0,1], and the residual energy of each sensor is randomly generated between 10 and 20 J. The moving speed of MC is 0.1 m/s, and the energy consumption rate of moving unit distance is 0.1 J/s. The rate at which MC charges the sensor is 1 J/s, and the time that the MC returns to the depot to charge itself is ignored here. The main simulation settings are provided in Table 4. In addition, the relevant data of the real-time energy consumption rate of the sensor are shown in Table 5.

After the network environment is initialized, we will conduct simulation training on the environment. Our implementation uses an experience replay buffer of $10^{5}$. The size of the minibatch is 1024. As for the neural networks, all networks (separate policies and those contained within the centralized critic networks) use a hidden dimension of 128, and the Leaky Rectified Linear units are used as the nonlinear activation. We train our models with the Adam optimizer [44] and set different initial learning rates when the network size is different. The key parameters used in the training stage are described in Table 6.

We have trained our model for three different environment settings on four NVIDIA GeForce GTX 2080ti for 10 h, after which the observed qualitative differences between the results of consecutive training iterations were ignored. We present one set of experimental results to describe the relationship between episodes and reward, which is shown in Figure 4. We can see that obtained rewards increase slowly through episodes to reach peak values after 240 training episodes. This is mainly caused by the efficient learning of AMADRL-JSSRC to the WRSN with dynamic energy changes so that agents could make reasonable decisions to obtain a greater reward.

Since the reward discount factor and penalty coefficient have a great impact on the performance of the algorithm in the training and testing process, we have made two sets of experiments, and the results are shown in Table 7 and Table 8, respectively. These experimental results prove that AMADRL-JSSRC will strive for a long-term reward rather than a short-sight reward when the reward discount factor approaches 1. Furthermore, with the increase in the penalty coefficient, the number of dead sensors gradually decreases. The reason is that in order to obtain a high global charging reward, AMADRL-JSSRC preferentially charges the sensors with low residual energy to avoid sensor death when a large value is assigned to ϵ. Therefore, in this paper, the penalty coefficient is set as 10, and the reward discount factor is set as 0.9.

### 4.2. Comparison Results against the Baselines

In this section, we compare the performance of the AMADRL-JSSRC with that of the ACRL algorithm, the GREEDY algorithm, the dynamic programming algorithm, and two typical online charging schemes algorithms NJNP and TSCA [16]. The detailed execution process of the above algorithms is shown in [31]. It is noted that some details of the baseline algorithms need to be adjusted. For example, we have replaced the reward calculation equation in line 13 of Algorithm 1, line 6 of Algorithm 2, and the line 10 of Algorithm 3 described in [31] with $r\left(S\left(k\right),a\left(k\right)\right)=\varpi^{d\left(k-1,k\right)}+\left(-\epsilon \right)N_{d}\left(k\right)$, and we change their seeking rule from the minimum global reward to the maximum global reward.

We consider three networks with different scales, including 50, 100, and 200 sensors; these environments are denoted as JSSRC50, JSSRC100, and JSSRC200. We have run our tests on WRSNs based on these environments, and the corresponding MC capacity is set as 50, 80, and 150 J. In addition, the base time of these three tests is set as 100 s, 200 s, and 300 s, respectively. Unless otherwise specified, these parameters are fixed during the test.

The tour length, the extra time, and the number of dead sensors obtained via different algorithms based on different JSSRC environments are shown in Table 9. It is observed that when the network size is small, such as the network with 50 sensors, the exact heuristic algorithm is better than AMADRL-JSSRC and ACRL algorithms in terms of average tour length and the average number of dead sensors. Meanwhile, the ACRL performance is slightly better than that of AMADRL-JSSRC at JSSRC 50. However, with the increase in network scale, the results of AMADRL-JSSRC and ACRL outperform the GREEDY, DP, JNJP, and TSCA significantly; the AMADRL-JSSRC and ACRL algorithms begin to show their superiority. The AMADRL-JSSRC algorithm is better than the ACRL algorithm, especially in the terms of the number of dead sensors. The reason for this phenomenon is that the charging ratio of the ACRL algorithm is fixed and cannot adjust adaptively according to the real-time charging demand, which will lead to some sensors becoming dead during the MC charging the selected sensors. The extra time comparisons are also presented in this table, where all the times are reported on one NVIDIA GeForce GTX 2080ti. We find that our proposed approach significantly improves the solution while only adding a small computational cost in runtime. Moreover, the extra time of AMADRL-JSSRC is longer than that of ACRL, verifying that multi-agent collaborative decision making consumes more computational cost.

## 5. Discussions

The impacts of the parameters on the charging performance, including the capacity of the sensor and the capacity of MC, and the performance comparison in terms of lifetime are discussed in this section. The test environment is set as 100 sensors, the baseline time is 300 s, the initial capacity of the sensor is 50 J, and the initial capacity of MC is 100 J. Meanwhile, the initial residual energy of sensors will change with the capacity of sensors. Since the baseline algorithms do not have the ability to adaptively control the charging ratio for each sensor, for a fair comparison, we introduce the optimal charging ratio from Table 4, which is named the charging coefficient in [31]. Therefore, the charging ratio of the baseline algorithms are ACRL ε=0.7, GREEDY ε=0.8, DP ε=0.9, NJNP ε=0.8, and TSCA ε=0.8, respectively.

### 5.1. The Impacts of the Capacity of the Sensor

As depicted in Figure 5, NJNP has the lowest tour length, and the average tour length gradually decreases with the increase in the capacity of sensors. The reason is that with the increase in the capacity of the sensor, the number of sensors’ charging requests will decrease on the premise of sufficient residual energy. Moreover, the charging for each sensor is also prolonged due to the increase in sensor capacity. Based on the fixed baseline time, the more time the MC spends on charging sensors, the less time it will spend on moving. Therefore, the average tour length decreases gradually. The fluctuation in Figure 5 is caused by the random distribution of sensor positions and the dynamic change of their energy consumption rate in each test experiment. Furthermore, the NJNP algorithm has the lowest tour length is because it preferentially charges the sensors close to MC. It is noted that the moving distance of AMADRL-JSSRC is slightly longer than that of ACRL. This is because AMADRL-JSSRC can determine different charging ratios for the selected sensors according to the real-time charging demand to avoid the punishment caused by the dead sensors. Therefore, AMADRL-JSSRC spends slightly less time on charging than ACRL. When the base time is fixed, more time will be spent on moving, resulting in a longer moving distance.

Figure 6 shows that with the increase in the capacity of the sensor, the average number of dead sensors shows an opposite change to the average tour length. Obviously, the average number of dead sensors of AMADRL-JSSRC is always the smallest. This is because the optimal charging ratios of the baseline algorithms are fixed, while the baseline algorithms are fixed. With the increase in the capacity of the sensor, the charging ratio for each selected sensor will be prolonged, increasing the risk of subsequent low residual energy sensor death. This result proves that the adaptive control of the charging ratio for each sensor could improve the charging performance effectively for the network.

### 5.2. The Impacts of the Capacity of MC

Figure 7 and Figure 8 show the impacts of the capacity of MC change on the average tour length and the average number of dead sensors, respectively. With the increase in MC capacity, when the baseline time is fixed, the MC could reduce the time of returning to the depot to charge itself. This change could shorten the moving distance of the MC and decrease the risk of subsequent low residual energy sensor death when it returns to or leaves the depot. Meanwhile, figures verify that compared with the ACRL algorithm, AMADRL-JSSRC gains a smaller number of dead sensors at the cost of increasing a certain moving distance.

### 5.3. Performance Comparison in Terms of Lifetime

We have analyzed the test results of six schemes under the fixed baseline lifetime. In this section, we explore the lifetime of the six schemes under different JSSRCs, which are JSSRC50, JSSRC100, and JSSRC200, until the termination condition is satisfied. These six algorithms have run 50 times independently, and the test results are shown in Figure 9, Figure 10 and Figure 11. It can be seen from the figure that although the fluctuation range of network lifetime obtained by the AMADRL-JSSRC algorithm is large, the lower and the upper bounds of network lifetime are still higher than the other five algorithms significantly. Moreover, with the increase in the number of sensors, this performance is outstanding significantly. It is noted that the network lifetime obtained by the AMADRL-JSSRC is better than the ACRL, which further proves that adjusting the charging ratio adaptively for each sensor could prolong the network lifetime effectively.

## 6. Conclusions

In this paper, a novel joint charging sequence scheduling and charging ratio control problem is studied, and an attention-shared multi-agent actor–critic-based deep reinforcement learning approach (AMADRL-JSSRC) is proposed, where a charging sequence scheduler and a charging ratio controller are employed to determine the target sensor and charging ratio by interacting with the environment. AMADRL-JSSRC trains decentralized policies in multi-agent environments, using a centralized computing critic network to share an attention mechanism, and it selects relevant policy information for each agent. Meanwhile, the AMADRL-JSSRC performance significantly prolongs the lifetime of the WRSN and minimizes the number of dead sensors, and the performance is more significant when dealing with large-scale WRSNs. In future work, the multi-agent reinforcement learning approach for multiple MCs to complete the charging tasks jointly is the key point for further study.

## Figures and Tables

**Figure 1 entropy-24-00965-f001:**
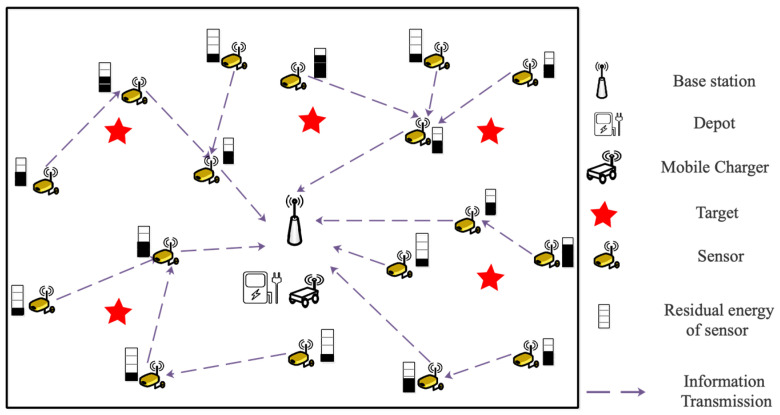
An example WRSN with a mobile charger.

**Figure 2 entropy-24-00965-f002:**
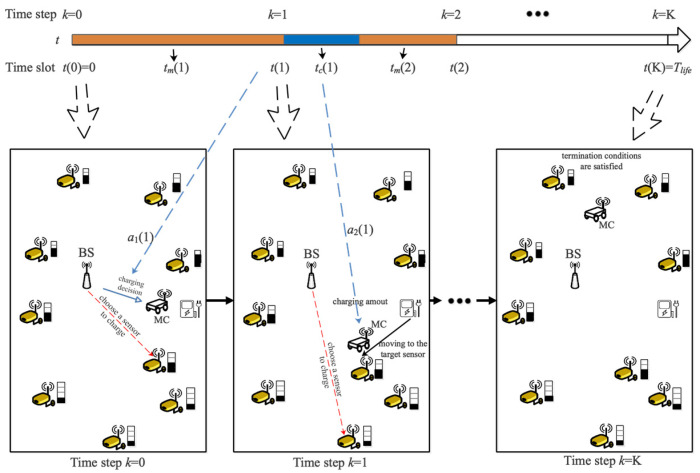
A scheduling example of JSSRC.

**Figure 3 entropy-24-00965-f003:**
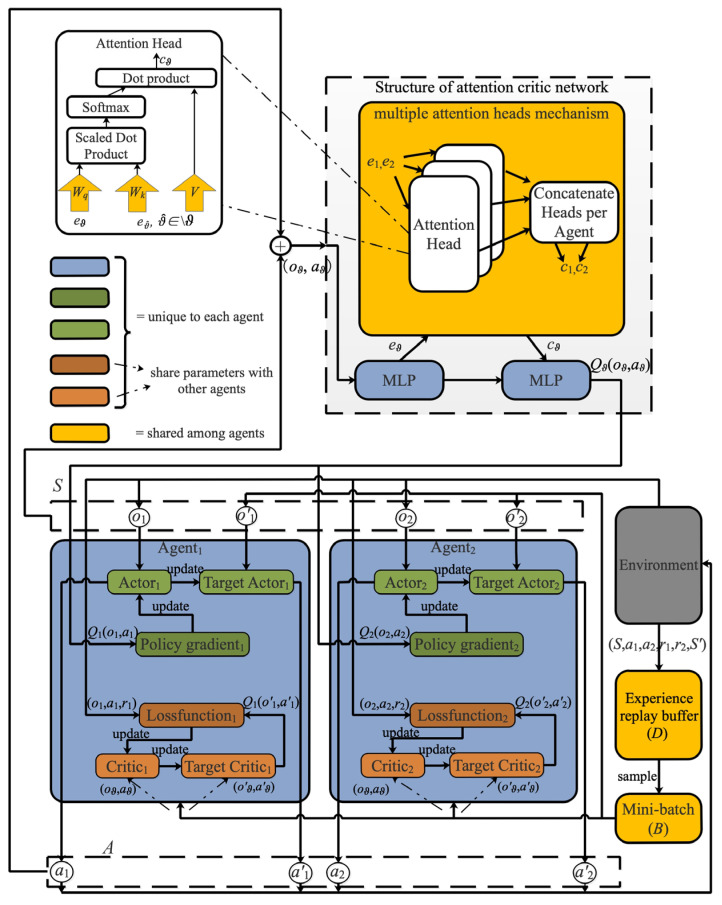
The structure of the AMADRL-JSSRC algorithm.

**Figure 4 entropy-24-00965-f004:**
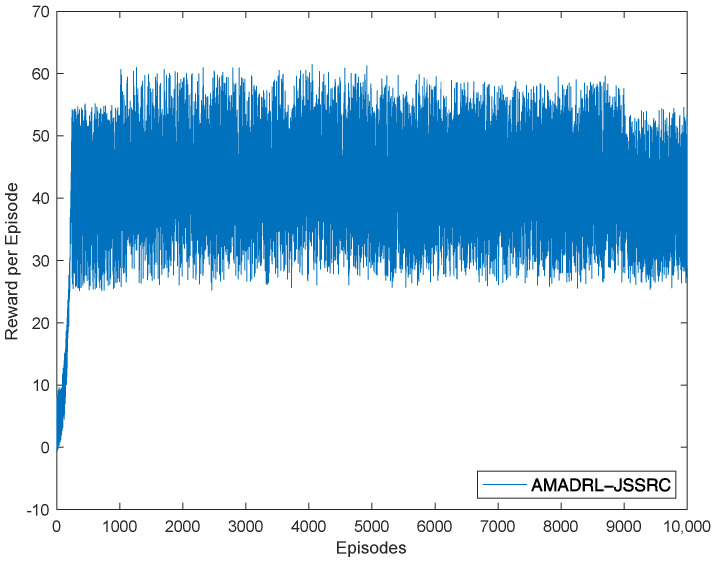
Reward per episode.

**Figure 5 entropy-24-00965-f005:**
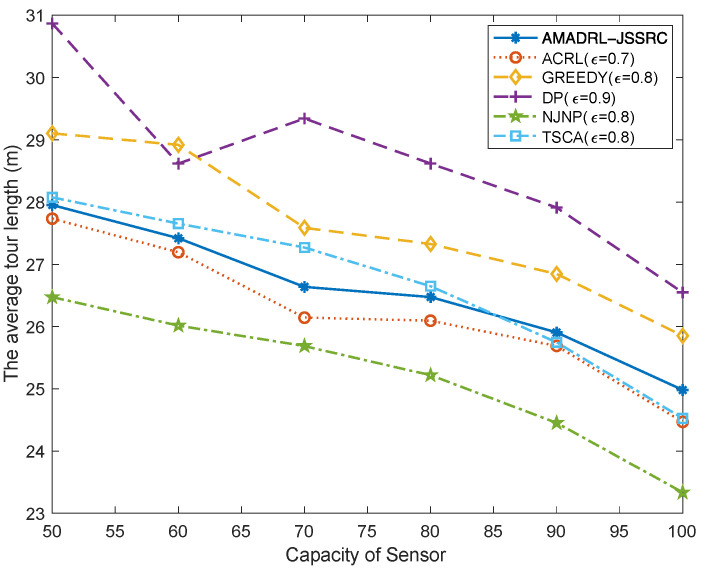
The impact of the capacity of the sensor on the average tour length.

**Figure 6 entropy-24-00965-f006:**
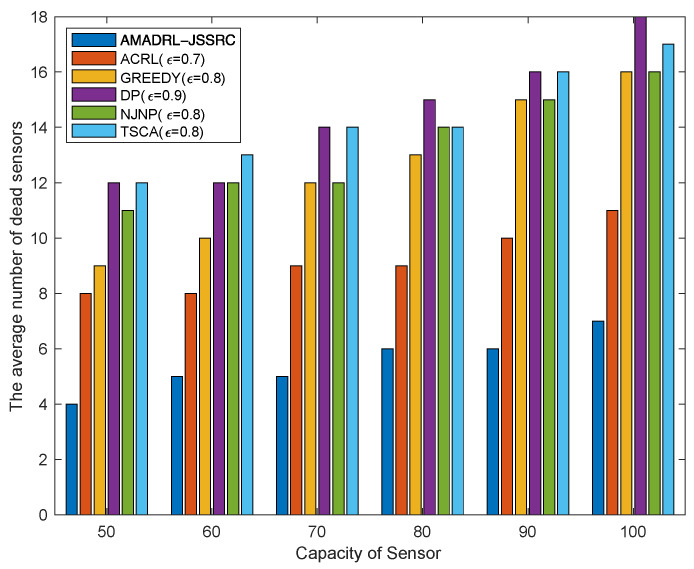
The impact of the capacity of the sensor on the average number of dead sensors.

**Figure 7 entropy-24-00965-f007:**
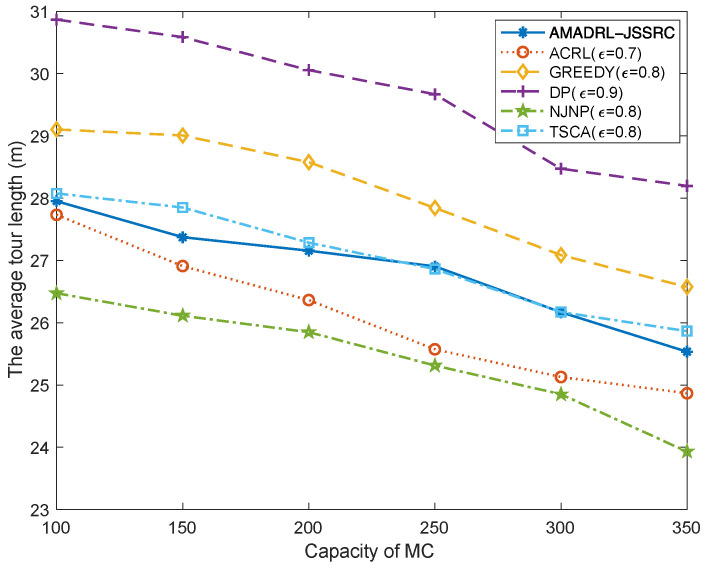
The impact of capacity of MC on the average tour length.

**Figure 8 entropy-24-00965-f008:**
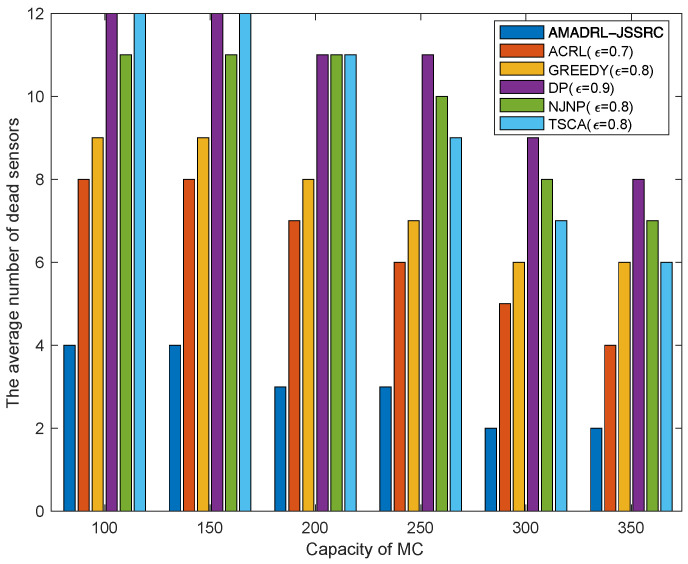
The impact of capacity of MC on the average number of dead sensors.

**Figure 9 entropy-24-00965-f009:**
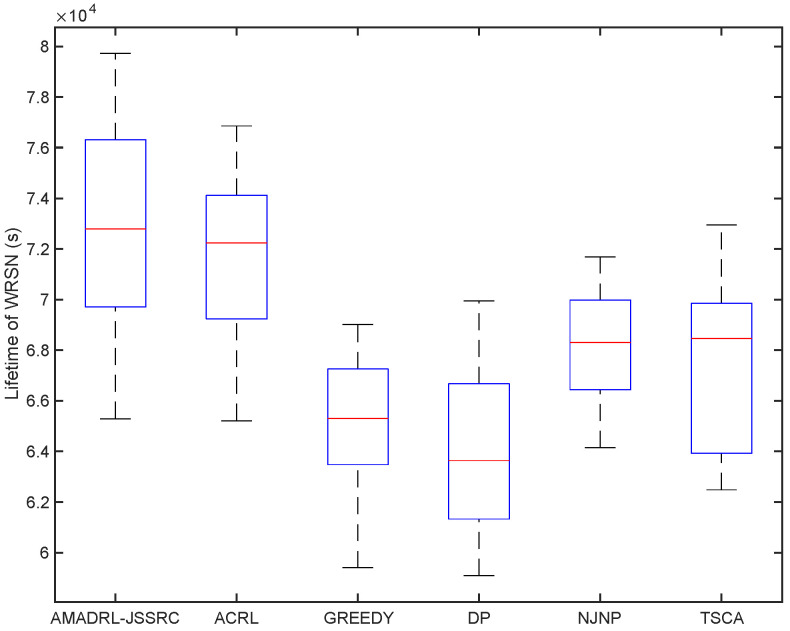
The lifetime of different algorithms on JSSRC50.

**Figure 10 entropy-24-00965-f010:**
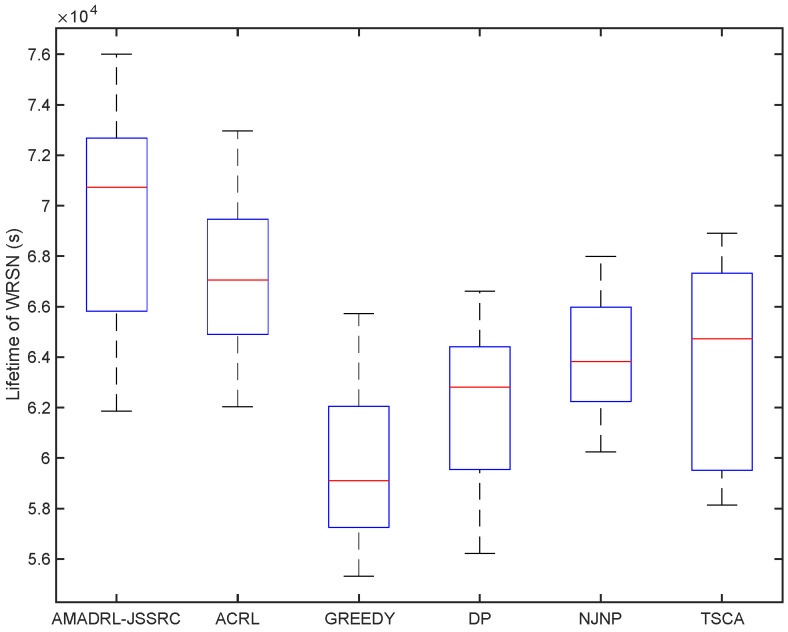
The lifetime of different algorithms on JSSRC100.

**Figure 11 entropy-24-00965-f011:**
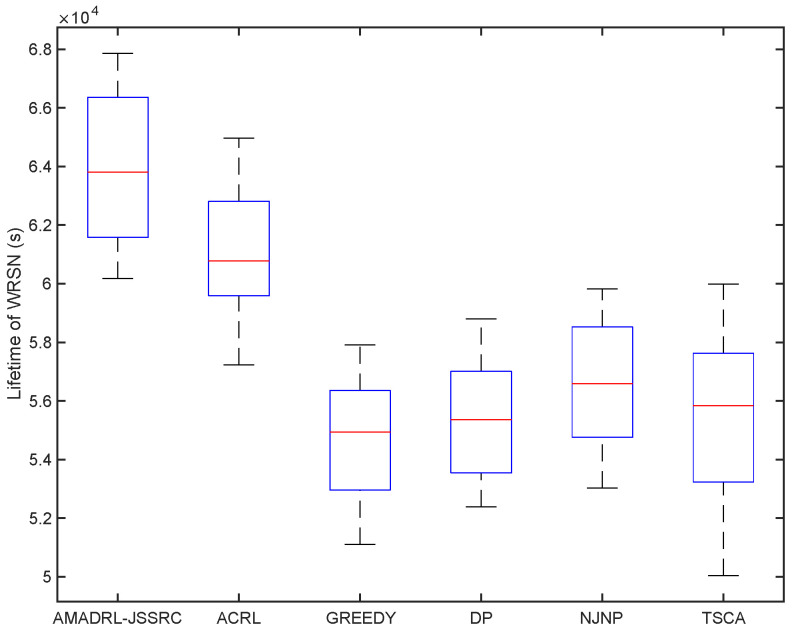
The lifetime of different algorithms on JSSRC200.

**Table 1 entropy-24-00965-t001:** Performance Comparison of the Existing Approaches and the Proposed Approach.

	Approach	Dynamic Change of the Sensor Energy Consumption	Charging Sequence Scheduling	Charging Ratio Control	Charging Sequence Scheduling and Charging Ratio Control Simultaneously
Off-line	[17]	No	Yes	No	No
	[18]	No	Yes	No	No
	[19]	No	Yes	No	No
	[20]	No	Yes	No	No
	[21]	No	Yes	No	No
On-line	[16]	Yes	Yes	No	No
	[25]	Yes	Yes	No	No
	[26]	Yes	Yes	No	No
	[27]	Yes	Yes	Yes	No
RL	[28]	No	Yes	No	No
	[29]	No	Yes	No	No
	[30]	No	Yes	No	No
	[31]	Yes	Yes	No	No
	Ours	**Yes**	**Yes**	**Yes**	**Yes**

**Table 2 entropy-24-00965-t002:** Abbreviations used in this paper.

Abbreviation	Description
WRSN	Wireless rechargeable sensor network
MC	Mobile charger
BS	Base station
Dis	Total moving distance of MC during the charging tour
JSSRC	Joint mobile charging sequence scheduling and charging ratio control problem
AMADRL	Attention-shared multi-agent actor–critic-based deep reinforcement learning
*S_mc_*	State information of MC
*S_net_*	State information of network
ACRL	Actor–critic reinforcement learning
DP	Dynamic programming
NJNP	Nearest-job-next with preemption
TSCA	Temporal–spatial real-time charging scheduling algorithm

**Table 3 entropy-24-00965-t003:** State Space Update.

Time Step	Observation	Agent 1		Agent 2		Immediate Rewards
		$\mathrm{Action}\;\mathrm{Space}\;(A_{1})$	Individual Reward $\left(r_{1}\right)$	$\mathrm{Action}\;\mathrm{Space}\;(A_{2})$	$\mathrm{Individual}\;\mathrm{Reward}\;(r_{2})$	
1	$O\left(1\right)=\left(o_{1}\left(1\right),o_{2}\left(1\right)\right)$	$a_{1}\left(1\right)$	$r_{1}\left(1\right)$	$a_{2}\left(1\right)$	$r_{2}\left(1\right)$	$R\left(1\right)=R(o_{1}\left(1\right),o_{2}\left(1\right),$ $a_{1}\left(1\right),a_{2}\left(1\right))$
…	…	…	…	…	…	…
*k*	$O\left(k\right)=\left(o_{1}\left(k\right),o_{2}\left(k\right)\right)$	$a_{1}\left(k\right)$	$r_{1}\left(k\right)$	$a_{2}\left(k\right)$	$r_{2}\left(k\right)$	$R\left(k\right)=R(o_{1}\left(k\right),o_{2}\left(k\right),$ $a_{1}\left(k\right),a_{2}\left(k\right))$
…	…	…	…	…	…	…
*K*	$O\left(K\right)=\left(o_{1}\left(K\right),o_{2}\left(K\right)\right)$	$a_{1}\left(K\right)$	$r_{1}\left(K\right)$	$a_{2}\left(K\right)$	$r_{2}\left(K\right)$	$R\left(K\right)=R(o_{1}\left(K\right),o_{2}\left(K\right),$ $a_{1}\left(K\right),a_{2}\left(K\right))$

**Table 4 entropy-24-00965-t004:** The Parameters of the Simulation Settings.

Parameter	Description	Value
	Network size	[0,1]×[0,1]
	Number of sensors	50–200
$E^{mc}$	MC initial energy	100 J
$v_{ms}$	Moving speed of MC	0.1 m/s
$e_{m}$	The speed of energy consumed on moving of MC	0.1 J/m
$P_{c}$	Charging speed of MC	1 J/s
ε	Charging ratio	
$E^{sn}$	Energy capacity of sensor	50 J
ω	The threshold of the number of dead sensors	0.5
$E^{0}$	The set of initial residual energy	10~20 J

**Table 5 entropy-24-00965-t005:** Energy parameters of sensor.

Parameter	Description	Value
$\xi_{1}$	Distance-free energy consumption index	$5\times 10^{-12\;}J/bit$
$\xi_{2}$	Distance-related energy consumption index	$1.3\times 10^{-4}\;J/bit$
ρ	Energy consumption for receiving or transmitting	$5\times 10^{-8}\;J/bit$
	Number of bits	$2\times 10^{4}$
r	Signal attenuation coefficient	4
	Per second packet generation probability	0.2~0.5

**Table 6 entropy-24-00965-t006:** Key Parameters of the Training Stage.

Parameter	Description	Value
*D*	Size of experience replay buffer	$10^{5}$
*B*	Size of mini-batch	1024
$\pi_{lr}$	Actor learning rate	5 × 10^−4^ (JSSRC50,100) 5 × 10^−5^ (JSSRC200)
$Q_{lr}$	Critic learning rate	5 × 10^−4^ (JSSRC50,100) 5 × 10^−5^ (JSSRC200)
$N_{p}$	Number of parallel environments	4
$N_{e}$	Number of episodes	$10^{4}$
$N_{pe}$	Number of steps per episode	100
$N_{cu}$	Number of critic updates	4
$N_{pu}$	Number of policy updates	4
$N_{m}$	Number of multiple attention heads	4
$N_{ut}$	Number of target updates	$10^{3}$
Adam	Optimizer method	
γ	Reward discount	0.9
ϖ	Reward coefficient	0.5
ϵ	Penalty coefficient.	10
τ	Update rate of target parameters	0.005
α	Temperature parameter	0.01

**Table 7 entropy-24-00965-t007:** Impact of the reward discount (ϵ=10).

γ	0.5	0.6	0.7	0.8	0.9
Reward	−60.72	−18.05	0.19	20.89	52.45
Number of dead sensors	25	18	13	8	5
Moving distance (m)	11.37	13.15	14.85	15.55	16.77

**Table 8 entropy-24-00965-t008:** Impact of the penalty coefficient (γ=0.9).

ϵ	0	1	5	8	10
Reward	60.33	38.29	37.64	40.12	52.45
Number of dead sensors	20	15	11	7	5
Moving distance (m)	14.54	15.09	15.88	16.23	16.77

**Table 9 entropy-24-00965-t009:** The Results Based on Different Algorithms Over Test Set.

Environment	Algorithm	Mean Length	Std	$\mathrm{Mean}\;N_{d}$	Base Time	Extra Time
JSSRC50	AMADRL-JSSRC	13.918	0.802	3	100	0.905
	ACRL	13.878	0.798	4	100	0.788
	GREEDY	13.902	0.834	2	100	0.647
	DP	14.068	0.856	6	100	0.743
	NJNP	13.834	0.815	5	100	0.516
	TSCA	14.028	0.755	4	100	0.498
JSSRC100	AMADRL-JSSRC	17.454	1.228	5	200	1.463
	ACRL	16.768	1.266	8	200	1.32
	GREEDY	18.233	1.445	13	200	1.38
	DP	18.088	1.328	13	200	1.12
	NJNP	16.891	1.306	12	200	0.995
	TSCA	17.718	1.205	11	200	0.936
JSSRC200	AMADRL-JSSRC	36.769	1.813	8	300	1.828
	ACRL	36.126	1.998	12	300	1.482
	GREEDY	37.856	3.162	19	300	1.635
	DP	37.532	2.376	18	300	1.864
	NJNP	35.513	2.265	17	300	1.465
	TSCA	35.921	2.169	16	300	1.416

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
