# Peer review of "Attention-Shared Multi-Agent Actor–Critic-Based Deep Reinforcement Learning Approach for Mobile Charging Dynamic Scheduling in Wireless Rechargeable Sensor Networks"

_entropy, 2022, doi:10.3390/e24070965_

Round 1
Reviewer 1 Report
The paper can be considered as a starting point for a future reseacth, but its actual contribution is poorly significant.
Reviewer 2 Report
This manuscript presents a new AMADRL-JSSRC algorithm to prolong the lifetime of wireless rechargeable sensor networks with a larger scale and reduce the number of death sensors based on th online control strategy. Overall, the manuscript is well prepared and the results are compared with baseline algorithms. Here are some minor suggestions:
(1)In introduction, authors listed and compared a dozen algorithms including offline and online strategy. The performances, drawbacks and improvements of each algorithms vs proposed algorithm should be summarized and highlighted clearly by a table.
(2)Section 4 is missing, it is suggested to move sections 5.1 and 5.2 to section 4 as experimental setup and results and move sections 5.3 and 5.4 to section 5 as discussion.
Reviewer 3 Report
This paper is well written with clear background research and problem formulation. Few minor comments:
1. The definitions could be highlighted or written in other formats.
2. The paper has used a lot of abbreviations, the reviewer suggests having a list of abbreviations.
3. Should "Dis" be defined within section 2.1?
